# Use of Dupilumab in Bullous Pemphigoid: Where Are We Now?

**DOI:** 10.3390/jcm11123367

**Published:** 2022-06-12

**Authors:** Roberto Russo, Niccolò Capurro, Emanuele Cozzani, Aurora Parodi

**Affiliations:** 1Department of Health Science (DISSAL), University of Genoa, 16132 Genoa, Italy; russoroberto@outlook.com (R.R.); capurro.niccolo@gmail.com (N.C.); emanuele.cozzani@unige.it (E.C.); 2Unit of Dermatology, San Martino Polyclinic Hospital, 16132 Genoa, Italy

**Keywords:** bullous pemphigoid, dupilumab, autoimmune blistering diseases

## Abstract

Bullous pemphigoid (BP) is the most frequent autoimmune subepidermal bullous disease. At present, the main treatment options are represented by corticosteroids and immunosuppressant drugs. Steroids often need to be administered in high doses, with subsequent adverse events and safety issues, as BP mainly affects elderly people. As dupilumab, a recombinant fully human IgG4 monoclonal antibody with binding specificity to human interleukin-4 receptor IL-4Rα has become paramount in the treatment of atopic dermatitis, its use in autoimmune bullous diseases has been theorized and it has been used to treat patients with BP. Dupilumab seems to be an effective and safe option to treat recalcitrant BP. Here, we report the results of a literature review on the use of dupilumab in BP, including a total of 30 treated patients in 9 papers.

Bullous pemphigoid (BP) is the most frequent autoimmune subepidermal bullous disease. At present, the main treatment options are represented by corticosteroids and immunosuppressant drugs. Steroids often need to be administered in high doses, with subsequent adverse events and safety issues, as BP mainly affects elderly people. In fact, overall mortality is significantly increased in BP, owing to both comorbidities and iatrogenic immunosuppression. Therefore, novel effective treatment options with fewer safety concerns are a matter of ongoing research [1].

As dupilumab, a recombinant fully human IgG4 monoclonal antibody with binding specificity to human interleukin-4 receptor IL-4Rα, has become paramount in the treatment of atopic dermatitis, and its use in autoimmune bullous diseases was theorized in 2019 [2]. In fact, dupilumab has been demonstrated to modulate chemokine-ligand 18, IL-4 and IL-13. These are Th2-related cytokines that show higher levels in BP patients (both in sera and in blister fluid) and play a role in the maintenance of Th2-type responses, which are thought to be involved in the loss of tolerance against BP180 [2]. From then on, reports of use of dupilumab in BP have been published. We performed research via the PubMed, Google Scholar, and clinicaltrials.gov databases using the search terms “dupilumab” and “pemphigoid”.

The first case report actually dates back to 2018: a patient with BP refractory to prednisone was successfully treated with dupilumab [3]. Later, Seidman et al. as well as Saleh et al. described patients with intractable BP who failed to respond to many different therapies but reached stable improvement when treated with dupilumab, understanding “improvement” symptoms as pruritus and blistering becoming well-controlled [4,5]. Abdat et al. published the first multicentric case series: of 13 patients, 9 achieved complete clinical response, where “satisfactory response” was defined by the authors as a “documented clinical improvement and patient desire to continue dupilumab.” In no more than 4 months, three showed partial response and only one patient failed to respond [6]. Additionally, a case of nivolumab-induced BP successfully treated with dupilumab has been reported [7]. Unfortunately, it was not clarified if nivolumab needed to be discontinued; if not, dupilumab would represent a safe and effective treatment to allow oncologic patients to continue nivolumab, even in the event of drug-induced BP. Zhang et al. retrospectively compared 8 BP patients treated with dupilumab plus azathioprine or methylprednisolone to 16 conventionally treated patients. Adding dupilumab resulted in the better control of disease progression and accelerated the tapering of glucocorticoids, while no adverse event related to dupilumab were recorded [8]. Liu et al. reported three patients with recalcitrant BP who significantly improved after treatment with dupilumab [9]. More recently, a very interesting report by Shan et al. described the case of a patient with vesicular BP concurrent with pulmonary tuberculosis which contraindicated a prolonged immunosuppressive therapy. Treatment with dupilumab resulted in the remission of vesicular BP, while pulmonary tuberculosis stayed well controlled [10]. Dupilumab was also proposed together with omalizumab as a successful combination therapy for BP; however, it is debatable that the patient was treated with two expensive biologic agents without an evident rationale, whereas monotherapy with dupilumab was not administered in this case [11]. Takamura and Teraki presented a case of a diabetic woman whose BP promptly responded to dupilumab, and also described the decrease in the proportion of circulating IL-4- and IL-13-producing CD4+ T cells (but not CD8+ T cells) after treatment, suggesting that dupilumab may exert its effect on BP by suppressing effector Th2 cells. However, the patient from this report was on gliptins, and it is unclear if they were discontinued as BP was diagnosed [12]. Table 1 summarizes the reports on use of dupilumab in BP.

A significant aspect that emerges from this review of various papers on the use of dupilumab in patients with BP as an implementing therapy is its high safety profile. In fact, several authors report the use of dupilumab in patients with multiple chronic diseases, including three cases of malignant neoplasms and one case of active tuberculosis (contraindicating other treatments), without substantial adverse events. The use of dupilumab was also shown to be optimally tolerated in patients receiving polypharmacotherapy either for previous health issues or for the treatment of BP. Two patients even took dupilumab concurrently with other biologic drugs, namely nivolumab and omalizumab, without significant interactions. Notably, none of the patients treated with dupilumab had been previously or concomitantly treated with rituximab. In almost all cases, there was clearance of disease after the use of dupilumab. Dupilumab was also effective on the side of symptoms, completely or greatly reversing pruritus in virtually all but one patient according to the reports. Dupilumab could play a relevant role not only in the management of severe pruritus, but also in the pathogenetic mechanisms of BP. In fact, in addition to circulating IgG, IgE are also recognized to be able to target BP180 and to trigger eosinophil and mast cell degranulation, and their serum levels seem to correlate with disease activity. IgE-mediated IL-4 and IL-13 production may be implicated in the upregulation of Th2 which seems to be the predominant immunological response in BP patients, suggesting Th2 cells’ involvement in the loss of tolerance against BP180 [13,14,15]. Indeed, Th2-related cytokines including IL-4 and IL-5 are overexpressed in BP lesional skin. Autoreactive Th2 cells are believed to play a dual role in BP: they stimulate B cells’ autoantibody production through the CD40-CD40L interplay and participate in the recruitment and activation of eosinophils, which may contribute to maintaining a Th2-type response via the production of IL-4, IL-5, and IL-13 [3,16,17].

For these reasons, we can hypothesize that the reduction in disease activity obtained in the cases reported so far may be related to the reduction in Th2-type responses induced by the inhibition of IL-4 and IL-13 signal transduction induced by dupilumab.

Certainly, dupilumab should not be intended as a first-choice treatment, in consideration of its high costs for healthcare systems, but as a rescue therapy for selected patients with recalcitrant BP.

Of course, randomized multicentric clinical studies are needed. If the results are confirmed, dupilumab might become another arrow in the quiver for the treatment of BP, as well as a safe option.

## Figures and Tables

**Table 1 jcm-11-03367-t001:** Reports of patients BP treated with dupilumab.

	Year of Publication	Number of Patients	Gender	Age (Years)	Patients’ Comorbidities	Concomitant Medication	Systemic BP Medication before Dupilumab (OVERALL)	Disease Duration before Dupilumab Initiation (Months)	Response to Dupilumab
Shan Y et al. [10]	2022	1	M	32	Tuberculosis	Isoniazid, rifampicin, and ethambutol	Oral corticosteroid	3	Disease clearance
Saleh M et al. [5]	2021	1	M	80	Not reported	Not reported	Prednisone 40 mg daily, doxycycline 100 mg twice daily, and niacinamide 500 mg three times per day, mycophenolate mofetil 1000 mg twice daily	Not reported	Marked improvement after 2 weeks, followed by complete resolution
Klepper EM et al. [7]	2021	1	F	79	Melanoma	Nivolumab, levothyroxine, hydrochlorothiazide /losartan, atorvastatin	Topical steroids, fexofenadine, dapsone	Unknown	Clearance of pruritus and BP lesion after 4 weeks
Zhang Y et al. [8]	2021	8	3 M, 5F	64.50 median (IQR 45.5–71.75)	Cardiovascular disease (N = 3) Neurologic disorders (N = 1) Hyperlipidemia (N = 1) Cancers (N = 2)	Not reported	Methylprednisolone (0.6 mg/kg/d) (N = 8) Azathioprine (2 mg/kg/d) (N = 8)	2 Median (N = 8)	Cessation of new BP lesions (8 days median time); Median BPDAI activity score: 34.25 (range: 19–75) at week 0 and 3.7 (range: 0–9) at week 2; Disease clearance in 62.5% of patients
Liu X et al. [9]	2021	3	1 M, 2 F	54 median (range 18)	Psychiatric Disorders (N = 1) HBV+ (N = 1) Hypertension, type 2 diabetes mellitus, stroke, arrhythmias with sustained atrial fibrillation, HBV+ (N = 1)	Not reported	Methylprednisolone 80 mg/d (N = 2) Prednisone (N = 1) IVIG (N = 2) Cyclophosphamide (N = 2) Cyclosporine (N = 2) Topical corticosteroids (N = 3)	17 median (N = 3)	Disease clearance (N = 2); Disease progression (N = 1)
Seyed Jafari SM et al. [11]	2021	1	M	70	Obesity, type 2 diabetes mellitus, hypertension	Not reported	Topical corticosteroids, Dapsone (up to 150 mg/day) Methotrexate 7.5 mg /week Mycophenolate-mofetil 2 g/d, Omalizumab	24	Disease clearance
Abdat R et al. [6]	2020	13	8 M, 5F	78 median (IQR 70.5–84.5)	Not reported	Not reported	None (N = 1) Prednisone(N = 3) Methotrexate (N = 1) Doxycycline (N = 1) Prednisone and Methotrexate (N = 2) Prednisolone, MTX, IVIG (N = 1) Prednisone, doxycycline, and niacinamide (N = 1) Prednisone, mycophenolate, doxycycline and niacinamide (N = 1) Rituximab, IVIG, doxycycline, nicotinamide, and azathioprine (N = 1) Prednisone, mycophenolate, rituximab, and IVIG (N = 1)	20.8 median (N = 12)	Disease Clearance (N = 7); Improvement in pruritus and clearance of BP lesions (N = 2); Improvement in pruritus but no clearance of BP lesions (N = 1); No improvement in pruritus and clearance of BP lesions (N = 1); Improvement of pruritus and BP lesions (N = 1); No improvement in pruritus or BP lesions (N = 1)
Seidman JS et al. [4]	2019	1	M	89	Type 2 diabetes mellitus	Metformin	Doxycycline 100 mg twice daily, nicotinamide 500 mg twice daily, mycophenolate mofetil 1000 mg twice daily (peak of 1500 mg twice daily), and prednisone 10 mg daily	24 (including disease flares)	Improvement of pruritus after 2 weeks, complete BP lesions resolution after 7 weeks
Kaye A et al. [3]	2018	1	M	80	*Mycobacterium tuberculosis* and hepatitis B core laboratory positivities	Not reported	Prednisone	0.3	Improvement in pruritus within a week after first injection; Resolution of all blisters after 3 months
Takamura S and Teraki Y [12]	2022	1	F	72	Type 2 diabetes mellitus	Inhibitors of dipeptidyl peptidase 4	minocycline with nicotinic acid amide	1.5	Resolution of pruritus in 2 weeks and blisters in 4 weeks

## Data Availability

The data presented in this study are openly available (see references list).

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
