# Peer review of "Use of Dupilumab in Bullous Pemphigoid: Where Are We Now?"

_jcm, 2022, doi:10.3390/jcm11123367_

Round 1

Reviewer 1 Report

Thanks for submitting this article which provides a very nice overview about the previously published reports of dupilumab use in bullous pemphigoid. The article is well-written and concise.

In my opinion, to enable publication as a review article, the authors should include more details about the potential pathophysiological rationale of this treatment (comparable to a previous paper by the authors: reference 2). Naturally, the information should be updated an understanding as of 2022. Moreover, I think the article would benefit from a figure illustrating the rationale of IL4/13 blockade in BP (interleukins, key players of the immune system and so on).

Please also adress some minor aspects as follows:

- please include information about the search strategy to enable the reader to reproduce the findings

- the abstract might include a total number of published cases thus far and the total number of reported responses and adverse events

- the term "response" and "improvement" is not satisfactorily defined in the text. How did the authors of the case report define clinical response? As absence of blistering, amelioration of pruritus, clearing of all lesions or even using BPDAI? The table is more useful in this regard but this should be mentioned in the text.

- I am not a native speaker but I suppose the term "assume" is incorrect in line 54-55

- is there any rationale for dual use of Omalizumab and dupilumab specifically in BP? The authors might include a statement in this regard.

- Could the high number of clinical responses with dupiliumab be attributable to publication bias?

- please include a statement regarding potential conflicts of interest, especially Sanofi? Please briefly adress arguments against wide use of dupilumab in BP patients (e.g. high costs for healthcare system)

- would you expect typical ocular side effects in BP patients or are these adverse events limited to atopic dermatitis patients?

Author Response

Thanks for submitting this article which provides a very nice overview about the previously published reports of dupilumab use in bullous pemphigoid. The article is well-written and concise.

In my opinion, to enable publication as a review article, the authors should include more details about the potential pathophysiological rationale of this treatment (comparable to a previous paper by the authors: reference 2). Naturally, the information should be updated an understanding as of 2022. Moreover, I think the article would benefit from a figure illustrating the rationale of IL4/13 blockade in BP (interleukins, key players of the immune system and so on).

Thanks to the Reviewer for their suggestions. Some details on the pathophysiological rationale was added in the text, as well as a figure.

Please also adress some minor aspects as follows:

- please include information about the search strategy to enable the reader to reproduce the findings

The search terms and databases were included in the text.

- the abstract might include a total number of published cases thus far and the total number of reported responses and adverse events

A sentence was added to the abstract accordingly.

- the term "response" and "improvement" is not satisfactorily defined in the text. How did the authors of the case report define clinical response? As absence of blistering, amelioration of pruritus, clearing of all lesions or even using BPDAI? The table is more useful in this regard but this should be mentioned in the text.

Thanks to the Reviewer for pointing at this. It was better clarified in the text.

- I am not a native speaker but I suppose the term "assume" is incorrect in line 54-55

We agree with the Reviewer. The word was changed.

- is there any rationale for dual use of Omalizumab and dupilumab specifically in BP? The authors might include a statement in this regard.

We included a sentence stating our opinion on the dual use of both biologics in BP. Thanks to the Reviewer for their suggestion.

- Could the high number of clinical responses with dupiliumab be attributable to publication bias?

It is reasonable that publication bias may have an impact on the data presented in literature. However, as use of biologics in BP, as well as off-label uses of dupilumab, are currently “hot topics”, in our opinion even negative results would have been reported. Plus, in the published case series, basically all patients somewhat improved with dupilumab, whereas if negative results were underreported, you would expect at least some failures.

- please include a statement regarding potential conflicts of interest, especially Sanofi? Please briefly adress arguments against wide use of dupilumab in BP patients (e.g. high costs for healthcare system)

We included both a statement excluding conflict of interests and a consideration about high cost of dupilumab if used widely. We agree they were necessary.

- would you expect typical ocular side effects in BP patients or are these adverse events limited to atopic dermatitis patients?

Pathogenic mechanisms of dupilumab-associated conjunctivitis are not completely known, however it is more frequent in patients with AD among other indications of dupilumab. It appears to be somehow AD-specific. In fact, published papers do not report an increased incidence of conjunctivitis in BP patients treated with dupilumab.

Reviewer 2 Report

The manuscript addresses a modern aspect in the treatment of bullous pemphigoid, namely dupilumab. The article aims to be a review. However, several aspects would improve the quality of the article, and would be in the benefit of the reader:

-expanding explanation for mechanism of action should be inserted

_methodology section with the exact criteria for study search and incusion in the narrative review, with keywords and MeSH terms, in order to increase reproductibility of the study and to minimise selection bias

-following article should be included (I mention that this article does not represent the work of the reviewer and there is no conflict of interests), as it adressed the same issue as the current manuscript:  Takamura S, Teraki Y. Treatment of bullous pemphigoid with dupilumab: Dupilumab exerts its effect by primarily suppressing T-helper 2 cytokines. J Dermatol. 2022 May 10. doi: 10.1111/1346-8138.16428. Epub ahead of print. PMID: 35538742.

Author Response

The manuscript addresses a modern aspect in the treatment of bullous pemphigoid, namely dupilumab. The article aims to be a review. However, several aspects would improve the quality of the article, and would be in the benefit of the reader:

-expanding explanation for mechanism of action should be inserted

_methodology section with the exact criteria for study search and incusion in the narrative review, with keywords and MeSH terms, in order to increase reproductibility of the study and to minimise selection bias

-following article should be included (I mention that this article does not represent the work of the reviewer and there is no conflict of interests), as it adressed the same issue as the current manuscript:  Takamura S, Teraki Y. Treatment of bullous pemphigoid with dupilumab: Dupilumab exerts its effect by primarily suppressing T-helper 2 cytokines. J Dermatol. 2022 May 10. doi: 10.1111/1346-8138.16428. Epub ahead of print. PMID: 35538742.

We sincerely thank the Reviewer for their suggestions. We added sentences about mechanism of action and methodology. We also included the article proposed, which was published after we prepared the first version of our manuscript.